# Wheat Flour-Based Edible Films: Effect of Gluten on the Rheological Properties, Structure, and Film Characteristics

**DOI:** 10.3390/ijms231911668

**Published:** 2022-10-01

**Authors:** Jing Wang, Xinyu Sun, Xingfeng Xu, Qingjie Sun, Man Li, Yanfei Wang, Fengwei Xie

**Affiliations:** 1College of Food Science and Engineering, Qingdao Agricultural University, Qingdao 266109, China; 2Qingdao Special Food Research Institute, Qingdao 266109, China; 3School of Engineering, Newcastle University, Newcastle upon Tyne NE1 7RU, UK

**Keywords:** edible packaging, biocomposite film, wheat flour, gluten, rheological properties

## Abstract

This work investigates the structure, rheological properties, and film performance of wheat flour hydrocolloids and their comparison with that of a wheat starch (WS)–gluten blend system. The incorporation of gluten could decrease inter-chain hydrogen bonding of starch, thereby reducing the viscosity and solid-like behavior of the film-forming solution and improving the frequency-dependence, but reducing the surface smoothness, compactness, water vapor barrier performance, and mechanical properties of the films. However, good compatibility between starch and gluten could improve the density of self-similar structure, the processability of the film-forming solution, and film performance. The films based on wheat flours showed a denser film structure, better mechanical properties, and thermal stability that was no worse than that based on WS–gluten blends. The knowledge gained from this study could provide guidance to the development of other flour-based edible packaging materials, thereby promoting energy conservation and environmental protection.

## 1. Introduction

Food packaging plays a crucial role in protecting food, enhancing food shelf life, and reducing food wastage [1,2]. Plastic materials, as traditional packaging, are facing being banned from use due to their non-degradability, biosafety issues, and environmental unfriendliness [3,4,5]. Edible packaging based on natural polymers, including polysaccharides and proteins, has become a sustainable solution and attracted extensive attention in recent years [6,7,8].

Starch is considered one of the most important biopolymers for edible packaging due to its low cost, high abundance, good transparency, and excellent barrier to oxygen [9,10]. However, the low water resistance and poor mechanical properties of pure starch films impede their wide applications [9]. To cope with the property limitations above, mixing starch with other natural polymers, such as protein and fat, has been recognized as one of the most cost-effective methods [11,12]. The combined formulations of starch and protein have been widely investigated for edible packaging since improvements were gained in the film properties of the composite matrices with respect to that of each pure biopolymer [13,14,15]. However, the improvement will be discounted since the morphology, processability, and final properties of polymer blends are dependent on the degree of compatibility, and most polymer blends are thermodynamically immiscible or incompatible on the molecular scale [16,17]. Therefore, the understanding of the compatibility, morphology, and rheological property is of great significance.

Flour, the natural blends of starch and protein with each component in the original state, has a much higher yield, is more widely available, consumes less energy for production, and shows greater component compatibility than individual pure components from the same agricultural source, such as starch and protein, which reduces the costs and improves film properties and competitiveness with traditional plastic packaging [18,19,20]. Different sources of flour with a higher starch content, including pea, rice, wheat, potato, pumpkin, and sage, have been used as film-forming materials for food packaging production [18,19,21,22,23,24]. Previous studies showed that flour-based films had similar physical properties or better manufacturing properties compared with purified starch-based and/or protein-based films [18,22].

Wheat is a food staple raw material for the majority of the population and one of the most important crops in the world [25]. Wheat flour mainly contains starch (78–82%) [26] and gluten (8–16%) [27] and is one of the most abundant flour sources used as thermoplastic materials for food packaging. Both wheat starch (WS) [28,29] and gluten [30,31,32] from wheat flours are widely explored in the production of edible films and coatings. However, most previous research studies focused on WS films, wheat gluten films, and the improvement of their properties [28,33,34]. The interactions between WS and gluten are mainly concerned with the effect of gluten on the pasting properties and nutritional functions of WS [35,36]. There has been no work carried out to compare the characteristics and properties of films produced from wheat flour and blends of WS and gluten.

In this study, wheat flour films and blend films of WS and gluten were prepared by casting. The rheological properties, structure characteristics, and film performance of film-forming systems were investigated. Our results largely highlight the structure and properties of edible films based on wheat flour and their comparison with those of WS–gluten blend films with the same protein contents, which would provide relevant information about component interactions that occurred in the polymer matrix.

## 2. Results and Discussion

### 2.1. Morphology of Film-Forming Matrix System

Figure 1 shows the morphologies of the film-forming matrices of pure WS, wheat flours, and WS–gluten blends with different protein contents under optical microscopy. It can be seen that after being dyed with iodine, the WS phase became dark, the protein phase in a light color, and thus two phases could be identified. Clearly, starch and protein phases were immiscible, with a typical “sea-island” structure present. For all the samples, WS (dark), which was the major component of the film-forming matrix (accounting for more than 87%), was presented to be a continuous phase, with the scattering of protein domains (bright). The number of bright domains (the protein phase) increased with increasing protein content for both the wheat flour system and WS–gluten blend system.

For samples with the same protein contents, the protein domains in the starch matrix for the WS–gluten blend system were bigger and more irregular than that for the wheat flour system, suggesting better compatibility between the two polymers in the wheat flour system. As we all know, in wheat grains, WS exists as granules embedded in the protein matrix. This bonding state of WS and protein did not change during the flour milling process since wheat flour is a mixture of aggregates of protein matrix embedding starch granules [37]. The good compatibility between WS and protein for the wheat flour system was ascribed to their original binding state in wheat flour as in wheat grains.

### 2.2. Rheological Properties of Film-Forming Matrices

#### 2.2.1. Steady Rheological Properties

Figure 2 shows the viscosity curves of the film-forming matrices of pure WS, wheat flours, and WS–gluten blends with different protein contents accompanying shear rate change. It can be seen that all samples showed an obvious pseudoplastic (shear-thinning) behavior, which could be inferred from the decreased viscosity with increasing shear rate. Under shear force, for most polymer solutions, a pseudoplastic behavior will present under a shear force due to the disentanglement and molecule rearrangement [38,39]. Compared with wheat flours and WS–gluten blends, pure WS showed a higher viscosity and a stronger shear-thinning behavior. The addition of gluten decreased the viscosity and pseudoplastic behavior of WS. With increasing protein content, the viscosity and pseudoplastic behavior of film-forming pastes decreased both for the wheat flour system and WS–gluten blend system, indicating the addition of gluten decreased the viscosity and pseudoplastic behavior of WS. Accounting for this, protein impeded the leaching of amylose from starch granules by forming complexes with starch molecules on the granule surface during the preparation of the film-forming solution and thus led to decreased viscosity and solid-like behavior of the film-forming solution [40]. Besides, with the same gluten contents, the viscosity of wheat flours was lower than that of WS–gluten blends, which may be due to the good compatibility between starch and gluten in flour with their original state.

The values of *n* (flow behavior index) and *K* (fluid consistency index) calculated for the film-forming matrices of pure WS, wheat flours, and WS–gluten blends with different gluten content are shown in Table 1. Newtonian fluids have *n* = 1, while pseudoplastic fluids have *n* < 1. Moreover, the greater deviation of *n* from 1 indicates a stronger pseudoplastic behavior of the fluid. All the samples are pseudoplastic fluids since their *n* values were less than 1. With a higher gluten content, a greater *n* and smaller *K* (proportional to viscosity) are displayed, implying that the addition of gluten can weaken the pseudoplastic behavior and reduce the viscosity of the WS paste. Compared with WS–gluten blends, wheat flour samples with the same protein contents have a greater *n* and smaller *K*, marking improved processability.

#### 2.2.2. Linear Viscoelastic Regions

The storage modulus (*G′*) and loss modulus (*G″*) profiles for all samples as a function of strain (0.01–100%) were shown in Figure 3. At low strains (below 10%), all the samples were gel (*G′* > *G″*). Besides, the viscoelastic region of WS showed the widest range, and that of the samples containing gluten was narrowed with increasing gluten content. The viscoelastic range for the wheat flour sample was smaller than that for the WS–gluten blend with the same gluten content due to the improved destructive effects of gluten on the intermolecular hydrogen bonding of starch. When the strain was higher than 2.5%, high-gluten flour (HF) showed a broad decrease in *G′*. Given this, the strain was set at 1% for the following tests.

#### 2.2.3. Viscoelastic Properties during Heating

The dynamic viscoelastic properties of WS, wheat flours, and WS–gluten blends with different gluten content are shown in Figure 4. It is seen that for WS, a cooling gel, *G′*, was higher than *G″*, and both *G′* and *G″* decreased with increasing temperature since at a higher temperature, starch chain interaction was weakened, leading to increased chain mobility [38]. All the samples containing gluten were gel (*G′* > *G″*), which may be ascribed to the dominant role of starch (the starch content was over 87%). The moduli (*G′* and *G″*) of the samples containing gluten were lower than that of WS. Regarding this, the inter-chain hydrogen bonding in WS was disrupted by the addition of gluten and led to a softer gel texture.

For starch–gluten blends and wheat flour samples, a higher gluten content decreased the moduli. However, with the same gluten contents, the starch–gluten blends showed higher moduli than the wheat flour samples, indicating a more uniform structure developed in the wheat flour samples. Given this, the original state of starch and gluten in wheat flour might have improved their compatibility and led to a higher weakening effect of gluten on the WS gel.

#### 2.2.4. Dynamic Mechanical Properties

Figure 5 shows the results of *G*′ and *G*″ as a function of frequency at 25 °C for the film-forming matrices of pure WS, wheat flours, and WS–gluten blends with different protein contents. It is seen that all the samples presented a typical solid-like behavior (*G*′ > *G*″). At higher frequency, most samples have higher *G*′ and *G*″, suggesting the materials were more solid-like.

Table 2 lists the values of *n*′, *n*″, *G*_0_′, and *G*_0_″ for the film-forming matrices of pure WS, wheat flours, and WS–gluten blends with different gluten content at 25 °C. For all samples, the slopes (*n*′) were close to 0, and *G*_0_′ was higher than *G*_0_″, confirming their solid-like behavior [41]. The *n*′ and *n*″ values of the samples containing gluten were higher than those of WS, implying that they behaved more like a liquid than WS. WS showed some degree of frequency dependence, and this dependence was more apparent in the gluten-containing samples.

For WS–gluten blends and wheat flour samples, the slope was increased with higher gluten content, indicating gluten reduced the solid-like behavior and increased frequency-dependence of WS. The *n*′ value of wheat flour samples was higher than that of WS–gluten blends, suggesting that the effect of gluten on the frequency-dependence of WS was more apparent in wheat flour samples. For WS–gluten blends and wheat flour samples, both *G*_0_′ and *G*_0_″ were decreased with an increasing content of gluten, which may be ascribed to the weakening effect of gluten on the viscoelasticity of WS.

### 2.3. Characteristics of Films

#### 2.3.1. Fractal Structure

Figure 6 shows the SAXS patterns with the fitted curves for the films of pure WS, wheat flours, and WS–gluten blends with different protein contents. It is seen that within a certain limit, a self-similar fractal structure was seen in all the samples. For all the samples, the values of *α* (Porod slope) were smaller than three, implying a smooth surface of the films. The fractal dimension (*D*) values for the films of pure WS, wheat flours, and WS–gluten blends with different protein contents were determined with methods reported before [42] and included in Table 3. For all the samples, the *D* value of pure WS was the largest. In addition, with increasing gluten content, the value of *D* decreased progressively, suggesting that the incorporation of gluten in starch led to a lower density of the self-similar structure. A similar peak at about −2.05 showed in all wheat flour films, corresponding to the complexation between starch and lipid.

For samples with the same gluten contents, the *D* values for wheat flour films were larger than that for starch–gluten blend ones. Given this, the better compatibility between starch and gluten in wheat flours led to a self-similar structure with a higher density. Similar results were shown in hydroxypropyl methylcellulose–hydroxypropyl starch blends [38,42].

#### 2.3.2. Microscopic Morphology of Film Surface

The film performance depends on the final structure of an edible film, which has been affected by the interactions between film components [43,44]. In turn, the microstructural analysis of the films, which provides relevant information about the arrangement of film components, helps us in understanding film properties [45].

Figure 7 includes the surface micrographs of the films of pure starch, wheat flours, and WS–gluten blends with different protein contents under SEM. It can be seen that all the wheat flour films had a relatively smooth surface without any cracks and holes, indicating excellent compatibility between film components. Similar results have been previously reported for wheat flour-based films [45,46]. With increasing protein content, the roughness of wheat flour film surface increased, possibly due to the denaturation of protein during starch gelatinization and better arrangement between starch and water during cooling. A similar result was shown in WS–gluten blend films with different protein contents.

Compared with wheat flour films, a rough surface with irregular pits was viewed in the WS–gluten blend films with the same protein contents, suggesting WS and protein in the blending system have a lower degree of compatibility than that in wheat flour. This result may be attributed to the fact that the original state of WS and gluten in wheat grains improved the interaction and compatibility between starch and gluten. These results are consistent with the results of morphology.

#### 2.3.3. Tensile Properties

The tensile strength (*σ*_t_), elongation at break (*ε*_b_), and elastic modulus (*E*) values of wheat flour and WS–gluten blend films with different protein contents are shown in Figure 8. The WS film showed lower *E* and *σ*_t_ than the films containing gluten, including wheat flour samples and WS–gluten blends, denoting that gluten reduced the rigidity of the WS film. Regarding this, a looser film structure might be formed due to the addition of gluten. This was confirmed by the SAXS results.

Compared with WS–gluten blends, wheat flour samples with the same gluten contents showed higher *E* and *σ*_t_, suggesting that wheat flour films have better mechanical properties. This corresponds to the better compatibility between WS and gluten in wheat flour since incompatible polymers tend to form weak points due to their phase-separated structure, which impedes stress transfer.

#### 2.3.4. Thermal Stability

Figure 9 shows the TGA and DTG (derivative thermogravimetric analysis) curves for various films. For the pure WS sample, there was a slight weight loss from 30 °C to 130 °C due to moisture evaporation, followed by a major weight loss in the temperature range of 180–350 °C related to the thermal decomposition of biopolymers. The thermal decomposition peak of WS for the pure WS film was seen at 309 °C, indicating that the WS film has good thermal stability. There was an absence of glycerol evaporation, although glycerol was also used in the preparation of the WS film, which may be due to the transesterification in the film preparation process [47]. However, the TGA curve for the gluten film showed three well-defined thermal degradation stages. The DTG peaks at 56 °C, 170 °C, and 306 °C are attributed to the volatility of water, glycerol, and thermal decomposition of gluten, respectively. All the samples containing gluten presented three thermal degradation stages. The DTG peak for glycerol evaporation shifted to a higher temperature, 220 °C. The DTG curves show that the thermal decomposition process of gluten merged into that of starch and peaked at 309 °C. With the same gluten contents, wheat flour films have similar thermal stability to WS–gluten blend ones.

#### 2.3.5. Water Vapor Permeability (WVP)

Table 4 lists the WVP for the films of pure WS, wheat flours, and WS–gluten blends with different protein contents (8.5%, 11.0%, and 12.2%). The WVP of the pure WS film was the lowest one among all films. With increasing gluten content, the WVP value of films increased both for the wheat flour system and WS–gluten blend system, indicating the addition of gluten decreased the water vapor barrier property of the WS film. The WVP of the film is the amount of water vapor passing through the film, which is related to the sorption and diffusivity of water vapor [22]. The addition of protein led to a much looser film structure, as shown by the SAXS results, and increased the diffusivity of water vapor in the film. Similar results have been reported in films based on a starch–whey protein isolate system [48].

Compared with WS–gluten blends, wheat flour samples with the same gluten contents showed higher WVP. When the protein content was low, the added protein molecules destroyed the network structure of WS and created additional pores [15]. Compared with WS–gluten blend films, wheat flour films with the same gluten contents created more additional pores, which was due to the better compatibility between WS and gluten (shown by the morphology results). This explains the higher WVP of wheat flour films than that of WS–gluten blend films.

#### 2.3.6. Fourier Transform Infrared Spectroscopy (FTIR)

Figure 10 shows the FTIR spectra for the films of pure WS, wheat flours, and WS–gluten blends with different protein contents (8.5%, 11.0%, and 12.2%). All the films exhibited a clear peak at about 1000 cm^−1^, which is the characteristic absorption peak of polysaccharide molecules, including the C–O tensile vibration and C–C tensile vibration [15]. The absorption peaks located at 1550 cm^−1^ and 1650 cm^−1^ correspond to the C=O and C=N groups in protein, respectively [49,50]. The band at 2800–3000 cm^−1^ is attributed to the stretching of the –C–H (CH_2_) group in the protein [50]. The band located in the ranges of 3100–3700 cm^−1^ is attributed to the basic stretching mode of the –OH group [47], which comes from the polysaccharide molecules in WS and some amino acids in protein. Generally, the formation of hydrogen bonds shifts this characteristic peak to a lower wavenumber position [51]. When gluten was incorporated in WS to prepare edible films, the –OH absorption peak of the film was observed to shift slightly to a high wavenumber, indicating a decrease in the inter-chain hydrogen bonding in WS. With increasing gluten content, the shift in the absorption peak of the films increased both for the wheat flour system and the WS–gluten blend system. With the same protein contents, wheat flour films had greater shifts of the absorption peak than the WS–gluten blend films, implying the disruption effect of protein on the inter-chain hydrogen bonds of WS in the wheat flour system was more apparent than that in the WS–gluten blend system.

## 3. Materials and Methods

### 3.1. Materials

A food-grade WS was purchased from Zhilongkai Co., Ltd. (Kaifeng, China). Gluten was purchased from Binzhou Zhongyu Food Co. Ltd. (Binzhou, China). Three kinds of wheat flour (LF as low gluten wheat flour, MF as medium gluten wheat flour, HF as high gluten wheat flour) with different protein contents (8.5%, 11.0%, and 12.2%) were respectively supplied by Weifang Kite Flour Co., Ltd. (Weifang, China), Yihai Kerry Arawana Holdings Co., Ltd. (Shanghai, China), Wudeli Flour Group Co., Ltd. (Handan, China), with specifications listed in Table 5. Glycerin was purchased from Sinopharm Chemical Reagent Co., Ltd. (Shanghai, China).

### 3.2. Preparation of Film-Forming Matrix and Film Casting

Film-forming matrices (5 wt%) of pure WS, wheat flour, and WS–gluten blend with different protein contents (8.5%, 11.0%, and 12.2%) were prepared with the same method reported before [42]. First, WS and gluten (dry powder) were mixed and then dispersed in water (25 °C) with stirring for 30 min. Then, to gelatinize WS completely, the mixtures were heated to 95 °C with stirring in a water bath and maintained for 1 h. Last, the solutions were cooled down to 25 °C for rheological measurements.

Different edible films were prepared with a 5 wt% total concentration film-forming paste added with 1.5% glycerin according to the method in our previous work [42]. The sample solutions were cooled down to 60 °C and maintained for 30 min. Then, solutions (25 g) were dispended on the plastic Petri dishes (15 cm diameter) to cast films. Afterward, films were dried at 37 °C before being peeled from the dishes. Before further characterization, all the films were equilibrated at 75% relative humidity (RH) for at least three days. Table 6 lists the thicknesses of the films.

### 3.3. Characterization of the Film-Forming Matrix and Films

#### 3.3.1. Rheological Measurements

The rheological measurements of the film-forming matrices were explored with the same methods as reported before [38]. A strain-controlled rheometer (MCR302, Anton Paar, Austria) with a parallel-plate geometry (50 mm diameter) was applied in this section. The gap was set at 1 mm for the measurements.

Steady rheological test (flow pattern) with a pre-shearing (100 s^−1^ for 300 s) was performed at 25 °C with shear rate in the range of 10^−2^–10^1^ s. The relationship between shear stress (*τ*) and shear rate ($\dot{\gamma }$) of a film-forming solution can be described by the equation:(1)$$\tau =K{\dot{\gamma }}^{n}$$
where *K* and *n* are the fluid consistency index and flow behavior index, respectively.

Strain sweep measurements (oscillatory pattern) were performed at 25 °C before dynamic rheological properties investigation to obtain the linear range of viscoelasticity. The strain varied from 0.01% to 100%, and the frequency was set at 1 Hz.

Temperature sweeps were performed from 25 °C to 95 °C at a frequency of 1 Hz. The strain was set at 1%, and the heating rate was set at 2 °C/min. To prevent moisture evaporation, silicone oil (a small amount) was applied to the periphery of the parallel plates.

Frequency sweeps were also carried out from 0.1 rad/s to 100 rad/s at 25 °C. The strain was set at 1%. The frequency-dependence of storage modulus (*G*′) and loss modulus (*G*″) can be shown in the power-law equations:(2)$$G^{\prime }=G_{0}^{'}\omega^{n^{\prime }}$$
(3)$$G^{\prime \prime }=G_{0}^{\prime \prime }\omega^{n^{\prime \prime }}$$
where, *n*′ and *G*_0_′ are the slope and intercept of log *G*′—log *ω*, respectively; *n*″ and *G*_0_″ are the slope and intercept of log *G*″—log *ω*, respectively.

#### 3.3.2. Microscopy Observation

A Nikon Eclipse TE 2000-U optical microscope was applied to image the morphology of wheat flour pastes and WS–gluten blends according to the method in previous work [42]. The polymer solution of a 3 wt% concentration was obtained with the same method presented in Section 3.2 and then cast on glass. WS was dyed with a 1% iodine alcohol solution, which was prepared by blending iodine (1 g) and a potassium iodide solution (10 g) in a volumetric flask (100 mL), followed by adding alcohol. Afterward, the films were dried at room temperature before imaging.

#### 3.3.3. Small-Angle X-ray Scattering (SAXS)

SAXS was used to explore the fractal structure of the films [38]. The balanced film was placed on the sample rack of the SAXS (Nano-inXider, Xenocs, Grenoble, France), and the measurement time of each sample was 2 h in order to reduce noise. With air as the background, 2D images were transformed into 1D curves using XSACT analysis software. The SAXS data in the angular range of 0.007 < *q* < 0.368 Å^−^^1^, where *q* = 4πsin*θ*/*λ*, in which 2*θ* is the scattering angle of the X-ray source. Before further analysis, the data of all samples were background-subtracted and normalized.

The fractal dimension (*D*) can be determined from the Porod slope α according to the Porod equation:(4)$$I\left(q\right)\propto q^{-\alpha }$$
where *I* is the SAXS intensity, *q* is the scattering vector, and *α* is an exponent called the Porod slope [42].

#### 3.3.4. Scanning Electron Microscopy (SEM)

According to the methods reported in previous work [44], films were cut and fixed on copper stubs and coated with gold. Afterward, the microstructure of film surfaces was observed at a magnification of 300× on a scanning electron microscope (Gemini 300, ZEISS, Jena, Germany) with an accelerating voltage of 5 kV.

#### 3.3.5. Mechanical Properties

The mechanical properties of the films were measured according to the ASTM D5938 standard using an Instron tensile testing apparatus (5943) [38]. Tensile strength, elongation at break, and elastic modulus were collected at a crosshead speed of 10 mm/min. Five replicate measurements were carried out for each sample.

#### 3.3.6. Thermal Stability of Films

The thermal stability of different films was evaluated using a thermogravimetric analyzer (TGA1, Mettler Toledo, Columbus, OH, USA) system [38]. The samples were heated from 35 °C to 600 °C with a rate of 10 °C/min in a nitrogen atmosphere.

#### 3.3.7. Water Vapor Permeability (WVP)

Water vapor permeability of the films (32.5 cm^2^) was measured by a water vapor transfer rate testing instrument (C360M, Labthink, Jinan, China) according to the GB 1037-88 standard.

#### 3.3.8. Fourier Transform Infrared Spectroscopy (FTIR)

FTIR spectra of different films were obtained using an FTIR spectrometer (Nicolet islo, Thermo, Waltham, MA, USA) to investigate possible interactions between the components of edible films. The spectra were recorded over a range of 500–4000 cm^−1^ at room temperature with an accumulation of 32 scans.

## 4. Conclusions

This study concerns the effect of gluten content on the structure, rheological properties, and film performance of edible films based on wheat flour. The content of gluten in the matrix influenced the rheological properties and film performance of starch. A higher gluten content led to a decreased viscosity, pseudoplasticity, solid-like behavior, gel strength, and increased frequency dependence. Regarding this, the incorporation of gluten in starch disrupted the inter-chain hydrogen bonding in starch and led to a softer gel texture. An increase in gluten content led to a rougher film surface and smaller compactness of the self-similar structure, as reflected by a higher fractal dimension. Gluten decreased the mechanical properties and the water vapor barrier performance of films.

The comparison of structure and properties were made between wheat flour hydrocolloids and the WS–gluten blend system. The compatibility between starch and gluten also plays an important role in the rheological properties, structure, and film performance of starch. Both morphology and SEM data confirmed the better compatibility of starch and gluten in wheat flour since they were in their original state as in wheat grains. This led to increased frequency dependence, improved processability, a denser self-similar structure, and improved mechanical properties. Therefore, flour-based edible films, with good processability, denser structure, and improved film performance, could be developed for food packaging.

## Figures and Tables

**Figure 1 ijms-23-11668-f001:**
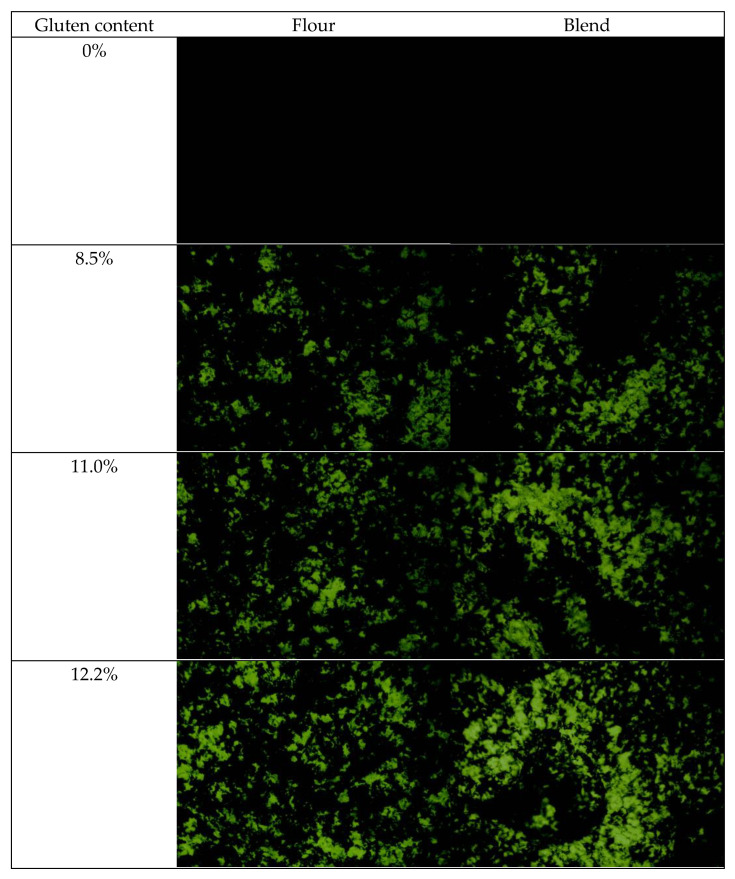
Light microscopy images (at a magnification of 20×) of the film-forming matrices of pure WS, wheat flours, and WS–gluten blends with different protein contents (8.5%, 11.0%, and 12.2%).

**Figure 2 ijms-23-11668-f002:**
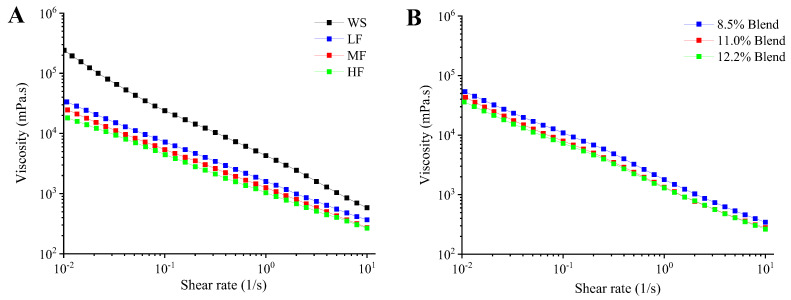
Viscosity as a function of shear rate for the film-forming matrices at 25 °C: (**A**) pure WS, wheat flours with different gluten contents (8.5%, 11.0%, and 12.2%), and (**B**) WS–gluten blends with different gluten contents (8.5%, 11.0%, and 12.2%). Low-gluten wheat flour, LF; medium-gluten wheat flour, MF; high-gluten wheat flour, HF; WS–gluten blend with 8.5% gluten content, 8.5% blend; WS–gluten blend with 11.0% gluten content, 11.0% blend; WS–gluten blend with 12.2% gluten content, 12.2% blend.

**Figure 3 ijms-23-11668-f003:**
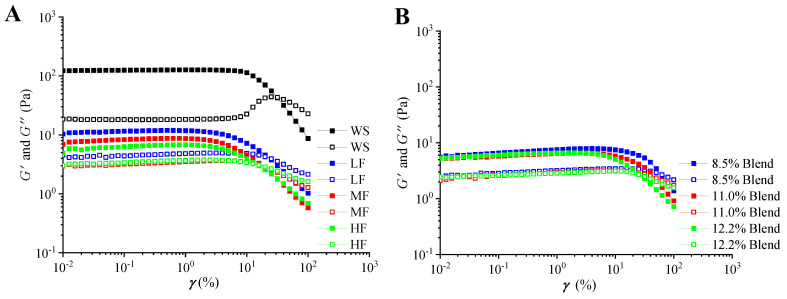
Storage modulus (*G′*, solid) and loss modulus (*G″*, hollow) as a function of strain of the film-forming matrices at a frequency of 1 Hz at 25 °C: (**A**) pure WS, wheat flours with different protein contents (8.5%, 11.0%, and 12.2%), and (**B**) WS–gluten blends with different protein contents (8.5%, 11.0%, and 12.2%).

**Figure 4 ijms-23-11668-f004:**
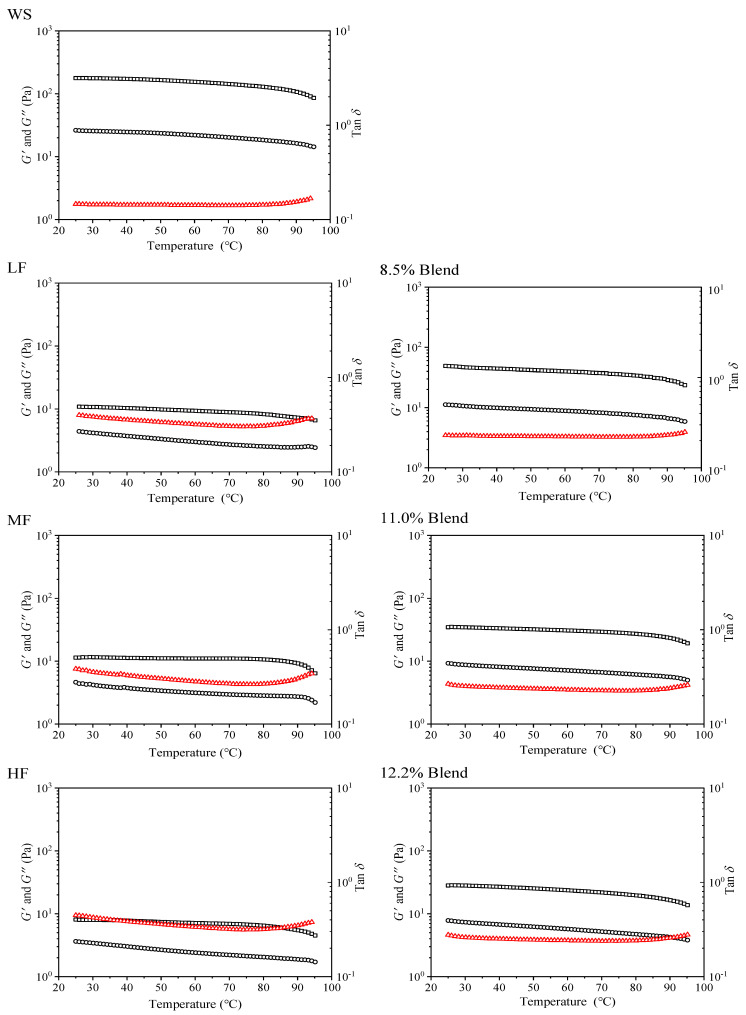
Storage modulus (*G′*, square), loss modulus (*G″*, circle), and tan (triangle) as a function of temperatures for the film-forming matrices of pure WS, wheat flours, and WS–gluten blends with different protein contents (8.5%, 11.0%, and 12.2%).

**Figure 5 ijms-23-11668-f005:**
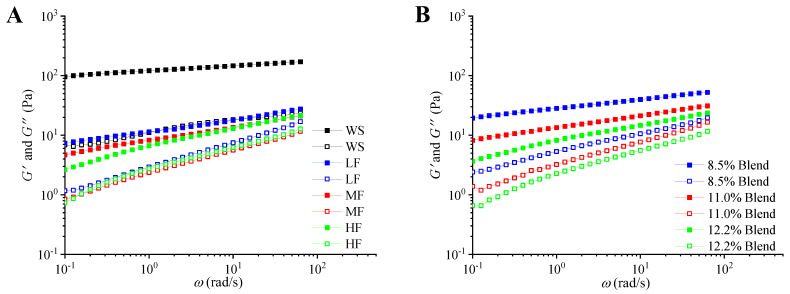
Storage modulus (*G′*, solid) and loss modulus (*G″*, hollow) as a function of frequency for the film-forming matrices at 25 °C: (**A**) pure WS, wheat flours with different protein contents (8.5%, 11.0%, and 12.2%), and (**B**) WS–gluten blends with different protein contents (8.5%, 11.0%, and 12.2%).

**Figure 6 ijms-23-11668-f006:**
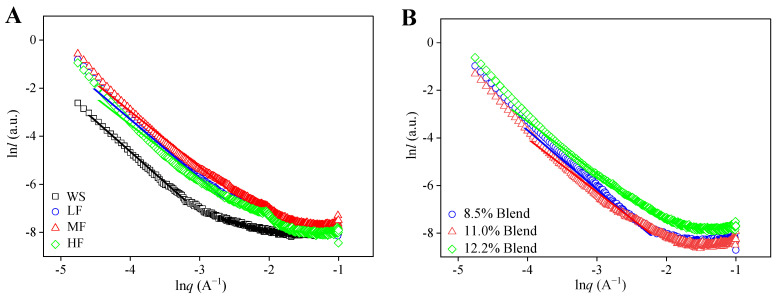
SAXS patterns and their fitted curves for the films: (**A**) pure WS, wheat flours with different protein contents (8.5%, 11.0%, and 12.2%), and (**B**) WS–gluten blends with different protein contents (8.5%, 11.0%, and 12.2%).

**Figure 7 ijms-23-11668-f007:**
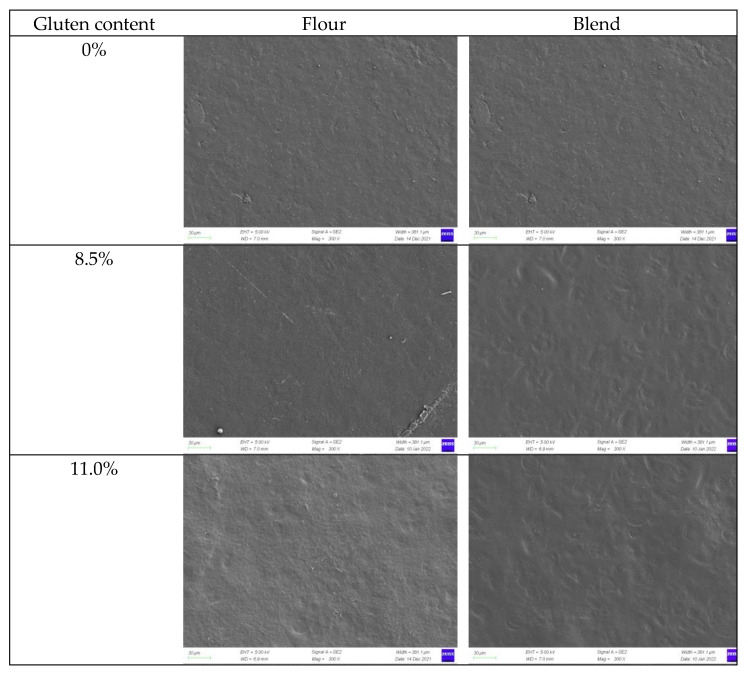
SEM images of the surface for the films of pure WS, wheat flours, and WS–gluten blends with different protein contents (8.5%, 11.0%, and 12.2%) at a magnification of 300×.

**Figure 8 ijms-23-11668-f008:**
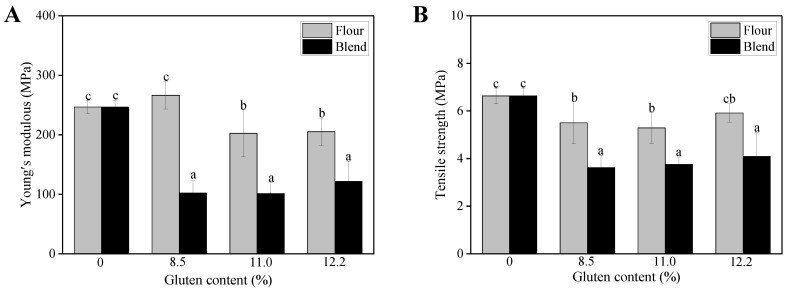
Tensile properties of the films of pure WS, wheat flours, and WS–gluten blends with different protein contents (8.5%, 11.0%, and 12.2%): (**A**) elastic modulus, (**B**) tensile strength, (**C**) elongation at break. Different letter (a, b, and c) means significant difference (*p* < 0.05).

**Figure 9 ijms-23-11668-f009:**
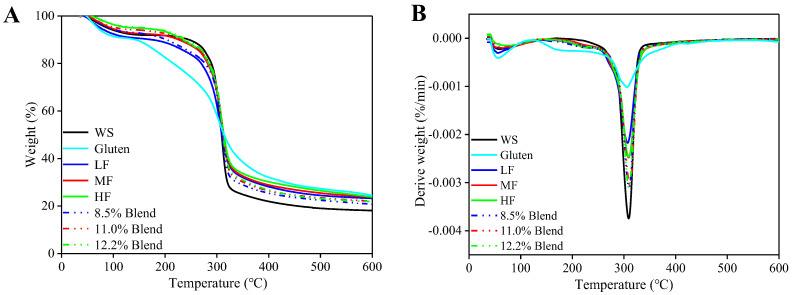
TGA curves (**A**) and DTG curves (**B**) for the films of pure WS, wheat flours, and WS–gluten blends with different protein contents (8.5%, 11.0%, and 12.2%).

**Figure 10 ijms-23-11668-f010:**
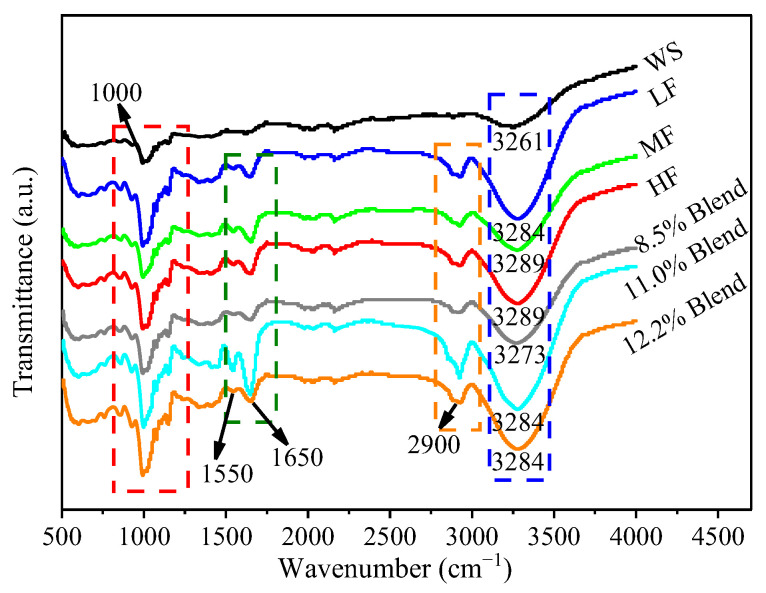
FTIR spectra for the films of pure WS, wheat flours, and WS–gluten blends with different protein contents (8.5%, 11.0%, and 12.2%).

**Table 1 ijms-23-11668-t001:** Flow behavior index (*n*), fluid consistency index (*K*) during increasing shear rate for the film-forming matrices of pure WS, wheat flours, and WS–gluten blends with different gluten content (8.5%, 11.0%, and 12.2%) at 25 °C.

Gluten Content (%)	Wheat Flours			WS–Gluten Blends		
	*n*	*K*	*R* ^2^	*n*	*K*	*R* ^2^
0	0.155 ± 0.012 ^a^	4.159 ± 0.244 ^e^	0.895	0.155 ± 0.012 ^a^	4.159 ± 0.244 ^e^	0.895
8.5	0.349 ± 0.007 ^d^	1.598 ± 0.036 ^c^	0.999	0.250 ± 0.010 ^b^	1.825 ± 0.031 ^d^	0.988
11.0	0.352 ± 0.014 ^d^	1.203 ± 0.055 ^ab^	1.000	0.264 ± 0.007 ^bc^	1.364 ± 0.031 ^b^	0.988
12.2	0.380 ± 0.014 ^e^	1.120 ± 0.055 ^a^	0.998	0.276 ± 0.003 ^c^	1.349 ± 0.045 ^b^	0.991

Note: Data are presented in the form of mean ± standard deviation. Different letters (a–e) mean significant difference (*p* < 0.05).

**Table 2 ijms-23-11668-t002:** Values of *n*′, *n*″, *G*_0_′, and *G*_0_″ for the film-forming matrices of pure WS, wheat flours, and WS–gluten blends with different gluten content (8.5%, 11.0%, and 12.2%) at 25 °C.

Sample	*n′*	*G* _0_ *′*	*R* ^2^	*n″*	*G_0_″*	*R* ^2^
WS	0.0837 ± 0.000 ^a^	117.695 ± 3.005 ^e^	0.9999	0.1760 ± 0.006 ^a^	12.948 ± 1.394 ^c^	0.9645
LF	0.2048 ± 0.011 ^c^	11.356 ± 0.094 ^bc^	0.9926	0.4223 ± 0.003 ^d^	2.895 ± 0.041 ^a^	0.9981
MF	0.2241 ± 0.002 ^d^	7.942 ± 0.312 ^ab^	0.9986	0.3772 ± 0.001 ^c^	2.312 ± 0.074 ^a^	0.9996
HF	0.2801 ± 0.006 ^f^	6.525 ± 0.175 ^a^	0.9987	0.3546 ± 0.004 ^b^	2.706 ± 0.021 ^a^	0.9992
8.5% Blend	0.1496 ± 0.003 ^b^	25.951 ± 2.342 ^d^	0.9977	0.3412 ± 0.015 ^b^	4.677 ± 0.431 ^b^	0.9969
11.0% Blend	0.2043 ± 0.002 ^c^	14.275 ± 0.643 ^c^	0.9972	0.3940 ± 0.008 ^c^	3.273 ± 0.121 ^a^	0.9964
12.2% Blend	0.2535 ± 0.000 ^e^	8.370 ± 0.148 ^ab^	0.9945	0.3922 ± 0.005 ^c^	2.245 ± 0.025 ^a^	0.9989

Note: Data are presented in the form of mean ± standard deviation. Different letter (a–e) means significant difference (*p* < 0.05).

**Table 3 ijms-23-11668-t003:** Fractal structure parameters for the films of pure WS, wheat flours, and WS–gluten blends with different protein contents (8.5%, 11.0%, and 12.2%).

Gluten Content (%)	Fractal Dimension (*D*)	
	Wheat Flours	WS–Gluten Blends
0	2.56	2.56
8.5	2.44	2.43
11.0	2.31	2.27
12.2	2.25	2.20

**Table 4 ijms-23-11668-t004:** Water vapor permeability for the films of pure WS, wheat flours, and WS–gluten blends with different protein contents (8.5%, 11.0%, and 12.2%).

Gluten Content (%)	WVP (g·cm/cm^2^·s·pa)	
	Wheat Flours	WS–Gluten Blends
0	1.647 ± 0.043 ^a^	1.647 ± 0.043 ^a^
8.5	1.738 ± 0.013 ^b^	1.687 ± 0.045 ^ab^
11.0	1.984 ± 0.043 ^d^	1.718 ± 0.023 ^b^
12.2	2.003 ± 0.052 ^d^	1.89 ± 0.007 ^c^

Note: Data are presented in the form of mean ± standard deviation. Different letter (a–d) means significant difference (*p* < 0.05).

**Table 5 ijms-23-11668-t005:** Specifications of wheat flours and gluten used in this work (obtained from the manufacturer).

Sample	Starch Content (wt%)	Gluten Content (wt%)	Fat Content (wt%)
LF	74.6	8.5	1.0
MF	73.5	11.0	1.6
HF	73.0	12.2	1.6
Gluten	12.5	80.6	0.8

**Table 6 ijms-23-11668-t006:** Thicknesses for the films of pure WS, wheat flours, and WS–gluten blends with different protein contents (8.5%, 11.0%, and 12.2%).

Gluten Content (%)	Thickness (mm)	
	Wheat Flour	WS–Gluten Blend
0	0.074 ± 0.002 ^a^	0.074 ± 0.002 ^a^
8.5	0.083 ± 0.003 ^b^	0.096 ± 0.001 ^d^
11.0	0.091 ± 0.004 ^c^	0.102 ± 0.003 ^e^
12.2	0.102 ± 0.002 ^e^	0.109 ± 0.002 ^f^

Note: Data are presented in the form of mean ± standard deviation. Different letter (a–f) means significant difference (*p* < 0.05).

## Data Availability

The data presented in this study are available on request from the authors.

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
