# Peer review of "Wheat Flour-Based Edible Films: Effect of Gluten on the Rheological Properties, Structure, and Film Characteristics"

_ijms, 2022, doi:10.3390/ijms231911668_

Round 1

Reviewer 1 Report

The manuscript “Wheat Flour-Based Edible Films: Effect of Gluten on the Rheological Properties, Structure and Film Characteristics” deals with the structure, rheological properties, and film performance of wheat flour hydrocolloids compared to a wheat starch-gluten blend system.

The manuscripts can be published in International Journal of Molecular Sciences after minor revisions:

Page 3 - line 95. Wheate starch, please correct. Wheat starch in all manuscript sometimes is abbreviated, sometimes not.

Page 8 - lines 200-201. “The fractal dimension (D) values for the films of pure starch, wheat flours and  starch-gluten blends with different protein content were determined with methods reported before [34]” . Please briefly describes the methods in Materials and Methods paragraph.

Page 8 - lines 211-212. “Similar results were shown in HPMC-HPS blends [31, 34]”.The meaning of abbreviation is difficult to understand.

Page 12- line 280, line 284 and line 286. “reported before [34]”, “methods as reported before [31]” and  “reported previous [34, 36]”. Should be better briefly describe the methods used for samples preparation and for characterization of matrices and films.

Page 12- line 295. “higher D”. It is better not use the abbreviation.

Author Response

Reviewer #1:

The manuscript “Wheat Flour-Based Edible Films: Effect of Gluten on the Rheological Properties, Structure and Film Characteristics” deals with the structure, rheological properties, and film performance of wheat flour hydrocolloids compared to a wheat starch-gluten blend system. The manuscripts can be published in International Journal of Molecular Sciences after minor revisions:

Response: Thanks for your careful review and positive comments. Below we have responded to your comments point by point, and related changes have been made in the manuscript.

  1. Page 3 - line 95. Wheate starch, please correct. Wheat starch in all manuscript sometimes is abbreviated, sometimes not.

Response: Thanks for your comment. Sorry for this misspelling. We have corrected it into “WS” and checked the whole manuscript to avoid misspellings. In order to make the manuscript more concise and uniform, we have applied “WS”, the abbreviation, instead of “wheat starch” after its first appearance. 

  1. Page 8 - lines 200-201. “The fractal dimension (D) values for the films of pure starch, wheat flours and  starch-gluten blends with different protein content were determined with methods reported before [34]” . Please briefly describes the methods in Materials and Methods paragraph.

Response: Thanks for your suggestions. We have added the calculation method of the fractal dimension (D) values in the Materials and Methods section (page 14, line 398-402).

“The fractal dimension (D) can be determined from the Porod slope α according to the Porod equation:

                                                                             (4)

where I is the SAXS intensity, q is the scattering vector and α is an exponent called the Porod slope [1].”

  1. Page 8 - lines 211-212. “Similar results were shown in HPMC-HPS blends [31, 34]”.The meaning of abbreviation is difficult to understand.

Response: Thanks for your comment. We abbreviated “hydroxypropyl methylcellulose” and “hydroxypropyl starch” to “HPMC” and “HPS” in our previous statement. To avoid confusion, we have changed HPMC-HPS” to “hydroxypropyl methylcellulose-hydroxypropyl starch”.

  1. Page 12- line 280, line 284 and line 286. “reported before [34]”, “methods as reported before [31]” and “reported previous [34, 36]”. Should be better briefly describe the methods used for samples preparation and for characterization of matrices and films.

Response: Thanks for your suggestions. We have added the description of the methods used in this work in sections 3.2 and 3.3 (page 13-15, line 337-425).

  1. Page 12- line 295. “higher D”. It is better not use the abbreviation.

Response: Thanks for your suggestions. To make the meaning clearer, we have changed the abbreviation “D” to “fractal dimension” (page 15, line 434).

  1. Wang, Y. F.; Wang, J.; Sun, Q. J.; Xu, X. F.; Li, M.; Xie, F. W., Hydroxypropyl methylcellulose hydrocolloid systems: Effect of hydroxypropy group content on the phase structure, rheological properties and film characteristics. Food Chem. 2022, 379.

Reviewer 2 Report

In this manuscript, Wang et al. presented the effect of gluten on the wheat flour based edible films. Based on the presented characterizations and explanations, I feel that this manuscript would fit better to a polymer related journal. Little to none molecular science is discussed in this work. Therefore, I would like to recommend a rejection of the manuscript and encourage the authors to submit this work elsewhere.

I have following comments for the authors to consider to improve their manuscript:     

- Abstract: “The incorporation of gluten could decrease inter-chain hydrogen bonding of starch, thereby reducing the viscosity and solid-like behavior of the film-forming solution” – This is used as an explanation a few times in section 2. Please provide sufficient background on this mechanism in the introduction section using appropriate references.  

- Researchers published a lot of in-depth work on the starch/gluten blend in recent years. Some examples are given below:

Food Hydrocolloids, Volume 113, April 2021, 106507

Food Sci. Technol 23 (2), Aug 2003, 264-269  

Agricultural Sciences in China, Volume 9, Issue 12, December 2010, Pages 1836-1844

J. Agric. Food Chem. 1999, 47, 2, 538–543

What is the novelty of this manuscript?

- “Few studies have been carried out about the comparison of the characteristics and properties of films produced from wheat flour and blends of wheat starch and gluten.” – Please provide references. How does this work differ from these few already carried out studies?

- “The good compatibility between wheat starch and protein for the wheat flour system was ascribed to their original state, which seems to be that in wheat grains.” – This explanation is not clear. Please rephrase and discuss more.  

- “the inter-chain hydrogen bonding in WS was disrupted by the addition of gluten and led to a softer gel texture.” – Please provide experimental proof to support this statement.   

- “For wheat flour samples and starch–gluten blends, a higher gluten content led to lower E and σt, denoting that gluten reduced the rigidity of WS film.” – However, it is very hard to establish a trend from the figure 8 data as “higher gluten content led to lower E and σt” mentioned here. What is the statistical significance for the presented data? Is there really any trend?  

- Figure 9: Please include gluten data in this study.

- “The samples were prepared with the same method reported before [34].” – Please briefly mention the film processing steps along with the final film properties like thickness and so on.

- “3.3. Characterization of the film-forming matrix and films” – Please provide the manufacturer and models of the instruments used for the characterization.

- Also, please mention what type of standard method (for example, ASTM) was followed for each characterization.  

- For packaging applications, it is very important to characterize the films to know their water resistance. Please include.  

Author Response

Reviewer #2:

In this manuscript, Wang et al. presented the effect of gluten on the wheat flour based edible films. Based on the presented characterizations and explanations, I feel that this manuscript would fit better to a polymer related journal. Little to none molecular science is discussed in this work. Therefore, I would like to recommend a rejection of the manuscript and encourage the authors to submit this work elsewhere. I have following comments for the authors to consider to improve their manuscript:    

Response: Thanks a lot for your careful review and comments. Actually, both wheat starch and protein, which were used for preparing edible films, are biomacromolecules and the interaction between these two molecules plays an important role in the film performance. Besides, we also expanded the discussion of the results in this work, especially from the perspective of molecular science. We have made significant changes to the manuscript according to your feedback and we believe the manuscript now meets the standard of the journal. Below we have responded to your comments point by point and related changes have been made in the manuscript.

  1. Abstract: “The incorporation of gluten could decrease inter-chain hydrogen bonding of starch, thereby reducing the viscosity and solid-like behavior of the film-forming solution” – This is used as an explanation a few times in section 2. Please provide sufficient background on this mechanism in the introduction section using appropriate references.

Response: Thanks for your suggestion. “The incorporation of gluten reducing the viscosity and solid-like behavior of the film-forming solution” are the results of the steady rheology test in our work. “The incorporation of gluten could decrease inter-chain hydrogen bonding of starch” is the proposed interpretation for this result.

Actually, we provide the advantage of polymer blending in the Introduction section that an improvement was gained in the film properties with respect to that of each pure polymer. What’s more, we emphasized that the morphology, processability and final properties of polymer blends are dependent on the degree of compatibility. The interpretation from the perspective of molecular science is that most polymer blends are thermodynamically immiscible or incompatible.

“To cope with the property limitations above, mixing starch with other natural polymers such as protein and fat has been recognized as one of the most cost-effective methods [1, 2]. The combined formulations of starch and protein have been widely investigated for edible packaging since an improvement was gained in the film properties of the composite matrices with respect to that of each pure biopolymer [3-5]. However, the improvement will be discounted since the morphology, processability and final properties of polymer blends are dependent on the degree of compatibility and most polymer blends are thermodynamically immiscible or incompatible on the molecular scale(Tanaka, Sako, Hiraoka, Yamaguchi, & Yamaguchi, 2020) [6, 7].”

To make the background clearer, we have added some sentences and discussions:

“Therefore, the understanding of the compatibility, morphology and rheological property is of great significance.” (page 1, line 44-45)

“The interpretation of this was that protein impeded the leaching of amylose from starch granules by forming complexes with starch molecules on the granule surface during the preparation of the film-forming solution and thus led to a decrease in viscosity and solid-like behavior of film-forming solution [8].” (page 4, line 108-112)

  1. Researchers published a lot of in-depth work on the starch/gluten blend in recent years. Some examples are given below:

Food Hydrocolloids, Volume 113, April 2021, 106507

Food Sci. Technol 23 (2), Aug 2003, 264-269 

Agricultural Sciences in China, Volume 9, Issue 12, December 2010, Pages 1836-1844

  1. Agric. Food Chem. 1999, 47, 2, 538–543

What is the novelty of this manuscript?

Response: Thanks for your comment. We have read the four papers you suggested carefully.

“Development and characterization of edible films based on gluten from semi-hard and soft Brazilian wheat flours (development of films based on gluten from wheat flours). Food Sci. Technol 2003, 23 (2), 264-269” and “Glass transition of wheat gluten plasticized with water, glycerol, or sorbitol. J. Agric. Food Chem. 1999, 47(2), 538-543” focused on the structure and properties of wheat gluten and its films.

“Effect of gluten on pasting properties of wheat starch. Agr. Sci. China, 2010, 9(12), 1836-1844” and “Mechanisms underlying the effect of gluten and its hydrolysates on in vitro enzymatic digestibility of wheat starch. Food Hydrocolloid.,  2021, 113, 106507 ” explored the interaction between wheat starch and gluten.

Indeed, these are excellent studies and the knowledge obtained from these works is useful. They mainly focus on the effect of gluten on the pasting properties and nutritional function of wheat starch. We have added the studies and references in the Introduction section (page2, line 59-64).

“Both wheat starch [9, 10] and gluten [11-13] from wheat flours are widely explored in the production of edible films and coatings. However, most previous research studies focused on WS films, wheat gluten films and their properties improvement [9, 14, 15]. The interactions between WS and gluten are mainly about the effect of gluten on the pasting properties and nutritional functions of WS [16, 17]. There has been no work carried out to compare the characteristics and properties of films produced from wheat flour and blends of WS and gluten.”

Our work focus on the structure-properties difference of edible films based on wheat flour and the blend of wheat starch-gluten blend and the knowledge obtained from this work would be vital to understand the mechanistic relationship between component compatibility and film performance.

  1. “Few studies have been carried out about the comparison of the characteristics and properties of films produced from wheat flour and blends of wheat starch and gluten.” Please provide references. How does this work differ from these few already carried out studies?

Response: Thanks for this comment. To our knowledge, works about the comparison of the characteristics and properties of films produced from wheat flour and blends of wheat starch and gluten have not been reported.

On page 2, line 64, we previously stated: “Few studies have been carried out about …”. To make this meaning more clear, we have changed “ there has been no work carried out to …”.

  1. “The good compatibility between wheat starch and protein for the wheat flour system was ascribed to their original state, which seems to be that in wheat grains.” This explanation is not clear. Please rephrase and discuss more.

Response: Thanks for your suggestion. We have rephrased our description of this message and added more discussion and related references to express it more clearly (page 2, line 87-92).

  1. “the inter-chain hydrogen bonding in WS was disrupted by the addition of gluten and led to a softer gel texture.” Please provide experimental proof to support this statement.

Response: Thanks for this suggestion. According your valuable suggestion, FTIR spectra of different films were carried out on an FTIR spectrometer (Nicolet islo, Thermo, America) to investigate possible interactions between the components of edible films. The results (Figure 10) and discussion were included in section 2.3.6 (page 12, line 303-325).

“2.3.6. Fourier Transform infrared spectroscopy (FTIR)

Figure 10. FTIR spectra for the films of pure WS, wheat flours, and WS–gluten blends with different protein contents (8.5%, 11.0%, and 12.2%).

Figure 10 shows the FTIR spectra for the films of pure WS, wheat flours, and WS–gluten blends with different protein contents (8.5%, 11.0%, and 12.2%). All the films exhibited a clear peak at about 1000 cm-1, which is the characteristic absorption peak of polysaccharide molecules including the C–O tensile vibration and C–C tensile vibration [5]. The absorption peaks located at 1550 cm-1 and 1650 cm-1 correspond to the C=O and C=N groups in protein, respectively [18, 19]. The band at 2800 – 3000 cm-1 is attributed to the stretching of the –C–H (CH2) group in the protein [19]. The band located in the ranges of 3100 – 3700 cm-1 is attributed to the basic stretching mode of the –OH group [20], which come from the polysaccharide molecules in WS and some amino acids in protein. Generally, the formation of hydrogen bonds shifts this characteristic peak to a lower wavenumber position [21]. When protein was incorporated in WS to prepare edible films, the –OH absorption peak of the film was observed to shift slightly to a high wavenumber, indicating a decrease in the inter-chain hydrogen bonding in WS. With increasing protein content, the shift in the absorption peak of the films increased both for the wheat flour system and the WS–protein blend system. With the same protein contents, wheat flour films had greater shifts of the absorption peak than the WS–protein blend films, implying the disruption effect of protein on the inter-chain hydrogen bonds of WS in the wheat flour system was more apparent than that in the WS–protein blend system.”

The results showed a difference in the absorption peak of films in the range of 3100 – 3700 cm-1, which correspond to the basic stretching of the –OH from the polysaccharide molecules in WS and some amino acids in protein. When gluten was incorporated into WS to prepare edible films, the –OH absorption peak of the film was observed to blue-shift slightly to a higher wavenumber, which proves that “the inter-chain hydrogen bonding in WS was disrupted by the addition of gluten”.

  1. “For wheat flour samples and starch–gluten blends, a higher gluten content led to lower E and σt, denoting that gluten reduced the rigidity of WS film.” However, it is very hard to establish a trend from the figure 8 data as “higher gluten content led to lower E and σt” mentioned here. What is the statistical significance for the presented data? Is there really any trend?

Response: Thanks for your valuable comments. We have added the statistical significance for the presented data in figure 8.

In section 2.3.3 (page 10, line 253-255), we previously stated: “For wheat flour samples and starch–gluten blends, a higher gluten content led to lower E and σt, denoting that gluten reduced the rigidity of WS film”. To avoid confusion, we have changed “The WS film showed lower E and σt than the films containing gluten including wheat flour samples and WS–gluten blends, denoting that …”

  1. Figure 9: Please include gluten data in this study.

Response: Thanks for this suggestion. According to your suggestion, we have added the result of gluten film and polished our discussion in section 2.3.4 (page 11, line 262-280). 

  1. 8. “The samples were prepared with the same method reported before [34].” Please briefly mention the film processing steps along with the final film properties like thickness and so on.

Response: Thanks for this suggestion. We have added the processing steps of the film-forming matrix and film in section 3.2 (page 13, line 337-350). We have also added the thickness data of all the films in Table 6 (page 13, line 351-354).

  1. 9. “3.3. Characterization of the film-forming matrix and films” Please provide the manufacturer and models of the instruments used for the characterization.

Response: Thanks for this suggestion. We have modified and perfected section 3.3 (page 13-15, line 355-425). The description of the methods for characterization of the film-forming matrix and films and the manufacturer and models of the instruments used in this work are all presented in this section. 

  1. Also, please mention what type of standard method (for example, ASTM) was followed for each characterization.

Response: Thanks for this suggestion. We have added the standard method for some characterization in section 3.3 (page 13-15, line 355-425). For example, the ASTM D5938 standard for mechanical properties, and the GB 1037-88 standard for water vapor permeability.

  1. For packaging applications, it is very important to characterize the films to know their water resistance. Please include.

Response: Thanks for this suggestion. According to your valuable suggestion, we carried out water permeability test. The WVP results of all films and the related discussions are shown in section 2.3.5 (page 11-12, line 281-302).

“2.3.5. Water vapor permeability (WVP)

Table 4 lists the WVP for the films of pure WS, wheat flours and WS -gluten blends with different protein content (8.5%, 11.0%, 12.2%). The WVP of the pure WS film was the lowest one among all films. With increasing protein content, the WVP value of films increased both for the wheat flour system and WS-protein composite system, indicating the addition of gluten decreased the water vapor barrier property of the WS film. The WVP of film is the amount of water vapor passing through the film, which is related to the sorption and diffusivity of water vapor [22]. The addition of protein led to a much more loose film structure as shown by the SAXS results and increased the diffusivity of water vapor in the film. Similar results have been reported in films based on starch-whey protein isolate system [23].

Table 4. Water vapor permeability for the films of pure WS, wheat flours and WS-gluten blends with different protein content (8.5%, 11.0%, 12.2%).

Gluten content (%)

WVP (g·cm/cm2·s·pa)

Wheat flours

WS-gluten blends

0

1.647±0.043a

1.647±0.043a

8.5

1.738±0.013b

1.687±0.045ab

11.0

1.984±0.043d

1.718±0.023b

12.2

2.003±0.052d

1.89±0.007c

Note: Data are presented in the form of mean ± standard deviation. Different letter (a, b, c and d) means significant difference (p < 0.05).

Compared with WS–gluten blends, wheat flour samples with the same gluten contents showed higher WVP. When the protein content was low, the added protein molecules destroyed the network structure of WS and created additional pores [5]. Compared with WS-gluten blend films, wheat flour films with the same gluten contents created more additional pores, which was due to the better compatibility between WS and protein (shown by the morphology results). This explains the higher WVP of wheat flour films than that of WS-gluten blend films.”

  1. da Silva, J. B.; Cook, M. T.; Bruschi, M. L., Thermoresponsive systems composed of poloxamer 407 and HPMC or NaCMC: mechanical, rheological and sol-gel transition analysis. Carbohyd. Polym. 2020, 240, 116268.
  2. Xu, Y. P.; Wang, T.; Shi, X., Enhanced dielectric and capacitive performance in polypropylene/poly (vinylidene fluoride) binary blends compatibilized with polydopamine. Mater. Design 2020, 195, 109004.
  3. Liu, C. L.; Yu, B.; Tao, H. T.; Liu, P. F.; Zhao, H. B.; Tan, C. P.; Cui, B., Effects of soy protein isolate on mechanical and hydrophobic properties of oxidized corn starch film. LWT-Food Sci. Technol. 2021, 147, 111529.
  4. Huntrakul, K.; Yoksan, R.; Sane, A.; Harnkarnsujarit, N., Effects of pea protein on properties of cassava starch edible films produced by blown-film extrusion for oil packaging. Food Packaging. Shelf. 2020, 24, 100480.
  5. Luo, S. M.; Chen, J. D.; He, J.; Li, H. S.; Jia, Q.; Hossen, M. A.; Dai, J. W.; Qin, W.; Liu, Y. W., Preparation of corn starch/rock bean protein edible film loaded with D-limonene particles and their application in glutinous rice cake preservation. IntJ. Biol. Macromol. 2022, 206, 313-324.
  6. Tanaka, Y.; Sako, T.; Hiraoka, T.; Yamaguchi, M.; Yamaguchi, M., Effect of morphology on shear viscosity for binary blends of polycarbonate and polystyrene. J. Appl. Polym. Sci. 2020, 137, (46), 49516.
  7. Ibrahim, B. A.; Karrer, M. K., Influence of Polymer Blending on Mechanical and Thermal Properties. Modern Applied Science 2010, 4, (9), 157-161.
  8. Juhani, O.; Rha, C., Gelatinisation of starch and wheat flour starch—A review. Food Chem. 1978, 3, (4), 293-317.
  9. Punia, S.; Sandhu, K. S.; Dhull, S. B.; Kaur, M., Dynamic, shear and pasting behaviour of native and octenyl succinic anhydride (OSA) modified wheat starch and their utilization in preparation of edible films. IntJ. Biol. Macromol. 2019, 133, 110-116.
  10. Bonilla, J.; Atares, L.; Vargas, M.; Chiralt, A., Properties of wheat starch film-forming dispersions and films as affected by chitosan addition. J. Food Eng. 2013, 114, (3), 303-312.
  11. Tanade-Palmu, P. S.; Grosso, C. R. F., Development and characterization of edible films based on gluten from semi-hard and soft Brazilian wheat flours (development of films of based on gluten from wheat flours). Food Sci. Technol. 2003, 23, (2), 264-269.
  12. Pouplin, M.; Redl, A.; Gontard, N., Glass transition of wheat gluten plasticized with water, glycerol, or sorbitol. J. Agric. Food Chem. 1999, 47, (2), 538-543.
  13. Chen, Y.; Li, Y. H.; Qin, S. L.; Han, S. Y.; Qi, H. S., Antimicrobial, UV blocking, water-resistant and degradable coatings and packaging films based on wheat gluten and lignocellulose for food preservation. Compos. Part B-Eng. 2022, 238, 109868.
  14. Song, X. Y.; Zuo, G. J.; Chen, F. S., Effect of essential oil and surfactant on the physical and antimicrobial properties of corn and wheat starch films. IntJ. Biol. Macromol. 2018, 107, 1302-1309.
  15. Dong, M. X.; Tian, L. J.; Li, J. Y.; Jia, J.; Dong, Y. F.; Tu, Y. G.; Liu, X. B.; Tan, C.; Duan, X., Improving physicochemical properties of edible wheat gluten protein films with proteins, polysaccharides and organic acid. LWT-Food Sci. Technol. 2022, 154, 112868.
  16. Chen, J. S.; Deng, Z. Y.; Peng, W. U.; Tian, J. C.; Xie, Q. G., Effect of gluten on pasting properties of wheat starch. Agr. Sci. China 2010, 9, (12), 1836-1844.
  17. Xu, H. B.; Zhou, J. P.; Yu, J. L.; Wang, S.; Wang, S. J., Mechanisms underlying the effect of gluten and its hydrolysates on in vitro enzymatic digestibility of wheat starch. Food Hydrocolloid. 2021, 113, 106507.
  18. Xiao, Q.; Woo, M. W.; Hu, J. W.; Xiong, H.; Zhao, Q., The role of heating time on the characteristics, functional properties and antioxidant activity of enzyme-hydrolyzed rice proteins-glucose Maillard reaction products. Food Biosci. 2021, 43, 101225.
  19. Tao, R.; Sedman, J.; Ismail, A., Characterization and in vitro antimicrobial study of soy protein isolate films incorporating carvacrol. Food Hydrocolloid. 2022, 122, 107091.
  20. Chen, X.; Cui, F. H.; Zi, H.; Zhou, Y. C.; Liu, H. S.; Xiao, J., Development and characterization of a hydroxypropyl starch/zein bilayer edible film. IntJ. Biol. Macromol. 2019, 141, 1175-1182.
  21. Furer, V. L.; Vandyukov, A. E.; Zaripov, S. R.; Solovieva, S. E.; Antipin, I. S.; Kovalenko, V. I., FT-IR and FT-Raman study of hydrogen bonding in p-alkylcalix 8 arenes. Vib. Spectrosc. 2018, 95, 38-43.
  22. Benincasa, P.; Dominici, F.; Bocci, L.; Governatori, C.; Panfili, I.; Tosti, G.; Torre, L.; Puglia, D., Relationships between wheat flour baking properties and tensile characteristics of derived thermoplastic films. Ind. Crop. Prod. 2017, 100, 138-145.
  23. Basiak, E.; Galus, S.; Lenart, A., Characterisation of composite edible films based on wheat starch and whey-protein isolate. Int. J. Food Sci.Tech. 2015, 50, (2), 372-380.

Round 2

Reviewer 2 Report

I feel that the authors made improvements to their manuscript based on the recommendations and it is now ready for acceptance.